# How Humic Acids Affect the Rheological and Transport Properties of Hydrogels

**DOI:** 10.3390/molecules24081545

**Published:** 2019-04-19

**Authors:** Martina Klucakova, Jiri Smilek, Petr Sedlacek

**Affiliations:** Materials Research Centre, Faculty of Chemistry, Brno University of Technology, Purkynova 118/464, 612 00 Brno, Czech Republic; smilek@fch.vutbr.cz (J.S.); sedlacek-p@fch.vutbr.cz (P.S.)

**Keywords:** humic acid, hydrogel, secondary structure, rheology, diffusion

## Abstract

Humic acids are often regarded as substances with a supramolecular structure which plays an important role in Nature. Their addition into hydrogels can affect their behavior and functioning in different applications. This work is focused on the properties of widely-used hydrogel based on agarose after addition of humic acids–the protonated H-form of humic acids and humic acids with methylated carboxylic groups. Hydrogels enriched by humic acids were studied in terms of their viscoelastic and transport properties. Rotational rheometry and methods employing diffusion cells were used in order to describe the influence of humic acids on the properties and behavior of hydrogels. From the point of view of rheology the addition of humic acids mainly affected the loss modulus corresponding to the relaxation of hydrogel connected with its flow. In the case of diffusion experiments, the transport of dyes (methylene blue and rhodamine) and metal ions (copper and nickel) through the hydrogel was affected by interactions between humic acids and the diffusion probes. The time lag in the hydrogel enriched by humic acids was prolonged for copper, methylene blue and rhodamine. In contrast, the presence of humic acids in hydrogel slightly increased the mobility of nickel. The strongest influence of the methylation of humic acids on diffusion was observed for methylene blue.

## 1. Introduction

Humic acids represent natural biological high-mineralized compounds originated from degraded organic matter as the product of animal and plant tissue decay [1,2,3]. Their structure is very complex and many research teams are focused on its investigation. Nevertheless, the concept of humic acids having a supramolecular structure is widely accepted. According to the supramolecular hypothesis [1,2,3,4,5,6], humic substances are associations of small molecules self-assembled by weak forces and hydrogen bonds. Such associations in solution are formed by the self-organization of hydrophobic and amphiphilic compounds and are isolated progressively from the water network [6]. These secondary structures of humic substances can be disrupted by the addition of organic acids [6,7,8,9] or metal ions [10]. Fischer [11] stated that humic supramolecular structures have polar surfaces and unpolar cores in soil. This means that highly polar subunits of humic substances are in contact with soil solution, that less polar subunits are in subsequent layers, and that unpolar subunits are inaccessible in the core of the humic supramolecular structure. Nebbioso and Piccolo [12] developed a “humeomics” approach to characterize the structure of humic substances based on sequential chemical fractionation. Other authors [9,13,14,15,16,17] have confirmed that supramolecular associations and macromolecules can co-exist in the structure of humic acids. 

Although, the structure of humic substances has been intensively studied, opinions are divided. On the one hand, it is clear that the conformational arrangement of humic substances can control their interactions with other components in Nature [1,10,12,18,19]; however, these processes are still not well understood, partially as a consequence of the ambiguous complex structure of humic substances themselves. The investigation of the secondary/supramolecular structure of humic substance is realized mainly in solutions. Our previous studies [20,21,22] based on the nontraditional methods of high resolution ultrasound spectrometry, dynamic light scattering, and micro-rheology showed that the molecular organization of humic acids in aqueous solutions could be divided according to three concentration ranges. Rearrangements were observed at concentrations of around 0.02 g·dm^−3^ and 1 g·dm^−3^. Changes in the measured values observed at around 0.02 g·dm^−3^ were less noticeable and were related to the formation of particles of between 100 and 1000 nm in size and changes in the hydration shells of humic particles resulting in changes in their elasticity. The “switch-over point” at around 1 g·dm^−3^ indicated changes in the secondary structure of humic acids connected with an increase in colloidal stability, a decrease in polydispersity, and minimum values of viscosity. The aggregation of humic particles and the formation of rigid structures in systems with concentrations higher than 1 g·dm^−3^ were detected [20,21,22]. Many other authors [5,23,24,25,26,27] observed a similar break in their results at around 1 g·dm^−3^ and considered it as a “switch-over point” in the secondary structure of humic acids, concluding that the apparent conformation of humic acids at higher concentrations might differ significantly from the conformation in more dilute systems. Christl et al. [25] stated that the high-concentration region should correspond to a pseudo-micellar organization phase. Our works also showed that solutions prepared by dilution had different properties than solutions prepared directly at the given concentration.

Humic substances are found in different environments (soil, water, coal, peat, sediments, etc.) and in various, mostly colloidal, forms (sol, suspension, gel) [28]. Humic sols can be expected to occur preferably in water systems and to have all or most of their functional or reactive groups free to interact. Humic gels might be found in water sediments, swelled peat, soil, or coal. Some of their functionalities are engaged in forming the gel structure but the whole system, including its interior, is still accessible to various interacting particles, penetrable through the gel network. In suspensions, humic constituents are tightly bonded in solid fraction, and essentially only the outer surface (including the surface of pores) is accessible to intermolecular interactions. In view of the above-described specific complex structure of humic substances and their common occurrence in gel form, hydrogel with incorporated humic acids was chosen Studies of humic hydrogels or hydrogels enriched by humic substances are relatively scarce. Vashurina et al. [29] added humic acids to starch-based hydrogels and achieved a technically important improvement in their rheological properties. Zielinska et al. [30] studied the partitioning of humic acids between solution and hydrogel. Two types of hydrogels were covered by humic solution, and the penetration of humic acids into hydrogels was monitored. A nonhomogeneous spatial distribution of humic acids in hydrogels and an accumulation of humic acids in a thin film on the hydrogel surface were observed. Chen et al. [31] prepared hydrogel beads from a starch-humic composite as a biodegradable adsorbent. Their adsorption capacity was higher than that of simple starch hydrogel. Kanmaz et al. [32] embedded humic acids into chitosan/poly(vinyl alcohol) hydrogel in order to develop a pH-sensitive hydrogel. They concluded that the swelling of hydrogel was controlled by the diffusion of solvent into pores and that the combination of humic acids with hydrogel had a synergic effect. Hydrogel containing humic acids exhibited smaller-sized open cells with interconnected narrow spaces, which made it more attractive for solvent diffusion. Ma et al. [33] assumed a double network in hydrogels based on polyacrylamide and humic substances. Its adsorption capacity and kinetics for different metal ions and their mixtures were studied. Several different binding sites in the hydrogel structure were identified. Hou et al. [34] prepared an injectable agarose hydrogel incorporating sodium humate and doxorubicin as an agent for tumor management based on the chemo-photo-thermal therapeutic effect. 

Our primal experiences [28,35,36,37,38,39,40] with humic hydrogel are connected with the development of methods concerning the reactivity and transport mapping of metal ions in natural systems containing humic substances. We believe that the transport properties and rates of interaction, either covalent or non-covalent, of humic substances, are as important as structural issues in evaluating and understanding the role of such substances in natural systems and human-driven applications. Transport processes, including diffusion and chemical interaction, are therefore important for understanding the complex behavior of natural systems and the role of their constituents. In our previous works [28,35,36,37,38,39,40], hydrogels were used as a model of natural systems containing humic substances. The hydrogels were prepared by means of the precipitation of humic acids dissolved in an alkaline solution by strong acid, followed by washing and centrifugation. They were physically bonded irreversible hydrogels with water contents of between 80 and 90% wt. It was found that the transport of metal ions was strongly affected by their interactions with humic acids [28,35,36,37,38,39,40]. In order to investigate the role of carboxylic groups in this process, we blocked them by methylation and investigated the transport of metal ions in hydrogels based on a mixture of methylated and non-methylated humic acids in different ratios [39,40]. In our recent studies, we focused on hydrogels based on agarose with the addition of humic acids, the rheological properties of hydrogels, and the transport of dyes in them [41,42,43]. Our previous results showed that a small amount of humic acids added into agarose hydrogel can significantly affect the transport of dyes in hydrogels in dependence on the humic content [41,43], temperature [41,42], pH, ionic strength [42], and the selective blocking of carboxylic functional groups of humic acids [43].

## 2. Results and Discussion

In this study, our previous findings were summarized and completed by conducting new experiments. The transport of metal ions and organic dyes through agarose hydrogel enriched by humic acids was studied alongside the rheological characterization of the used hydrogels. Native isolated humic acids and their methylated form were studied in order to investigate the influence of humic carboxylic groups on the diffusivity of metals and dyes, and the viscoelastic properties of hydrogels. Two metal ions–copper(II) and nickel(II)—and two organic dyes—methylene blue and rhodamine—were chosen as the diffusion probes, because of their different affinities to humic acids.

### 2.1. Diffusion Experiments

The method employing diffusion cells was used in our previous studies [41,43] and optimized for diffusion through agarose hydrogel containing humic acids as the reactive/immobilization agent. Concentration changes in both (donor and acceptor) compartments were monitored over time. Examples of experimental data obtained for the diffusion of Cu(II) ions through the hydrogel enriched by MHA are shown in Figure 1. In the first period, the diffusion probe was transported through the hydrogel and no changes were observed in the acceptor compartment. After that, the probe started to appear in the acceptor compartment and the initially zero concentration (*c*_A0_) gradually increased. In contrast, the concentration decrease in the donor compartment was slower due to the relatively high initial concentration of the diffusion probe (*c*_D0_). The first period of the experiment can be characterized by the time lag (*t*_L_), which is proportional to the hydrogel thickness (*L*) and the apparent diffusion coefficient (*D*_L_) according to Equation (1) [41,43,44]:*t*_L_ = *L*^2^/6*D*_L_(1)

It is assumed that the probes used penetrated only into the pores of hydrogels filled by solution and that their diffusion into the (solid) hydrogel network was negligible (practically equal to zero). The diffusion rate is thus dependent mainly on the shape and real length of the diffusion pathway. If the hydrogel contains an active substance able to interact with the probe, the transport through the hydrogel is affected by the interactions and a proportion of the probe particles can be immobilized by the interaction while passing through the hydrogel. The apparent diffusion coefficient *D*_L_ thus characterizes the transport through the hydrogel with an initially zero concentration of the probe. This means that the passing probe front is in contact with the hydrogel containing non-occupied binding sites up to the moment when it arrives at the end of the hydrogel layer and starts to permeate into the acceptor compartment. 

An overview of time lags and corresponding values of *D*_L_ is presented in Table 1, Table 2, Table 3 and Table 4. A prolongation of lag time for hydrogels enriched by HA and MHA in comparison with pure AG hydrogel was observed for all diffusion probes except for Ni(II) ions (their diffusion was slightly faster in enriched hydrogels). The time lags were much longer for organic dyes because of their sizes. Similarly, the values of *D*_L_ determined by means of Equation (1) decreased with the prolongation of the probe’s passage through hydrogels (Table 1, Table 2, Table 3 and Table 4). The highest values of *D*_L_ were obtained for metal ions. Only small differences were observed in the case of nickel which corresponded with its very low affinity to humic acids [36].

In general, when a diffusion probe passes through a hydrogel into the acceptor compartment and the probe is thus distributed throughout the whole hydrogel layer, the character of diffusion process changes. The transport of the probe from the donor to acceptor compartments through hydrogel still proceeds, but the hydrogel pores are filled by the probe solution and an equilibrium between freely mobile probe material and immobilized probe material can be assumed (i.e., if the hydrogel contains an active substance and the probe can react with its binding sites). The effective diffusion coefficient *D*_E_ characterizing this period of diffusion can be determined as [41,45]:−*βD*_E_*t* = ln [(*c*_D_ − *c*_A_)/(*c*_D0_ − *c*_A0_)](2)
where *β* is the so-called cell constant, i.e., the coefficient characterizing the geometrical parameters of the experimental apparatus. Its value was determined in our previous work [41] and used in this study for the calculation of *D*_E_. The concentrations *c*_D_ (donor compartment) and *c*_A_ (acceptor compartment) belong to a given time *t*; the initial concentrations are *c*_D0_ and *c*_A0_. The values of *D*_E_, presented in Table 1, Table 2, Table 3 and Table 4, were lower in comparison with the diffusion coefficients *D*_L_ determined on the basis of the time lag. The diffusion coefficients of dyes (*D*_E_ and *D*_L_) in AG hydrogels were of the same order of magnitude as values published in [46,47]. The effective diffusion coefficients *D*_E_ decreased when AG hydrogels were enriched by HA and MHA, but the differences were again small in the case of nickel. The differences between the results obtained for pure AG hydrogels and hydrogels enriched by humic substances can be the result of two effects. The first is a possible change in hydrogel structure. As described above, the structure of humic acids can be characterized by a supramolecular arrangement of relatively small particles, e.g., [6,8,11,12,13], often in co-existence with bigger macromolecules [9,13,14,15,16,17]. The structure of humic acids is very dynamic and sensitive to circumstances (concentration, pH, ionic strength) [21,22]. Therefore, their incorporation into AG hydrogel can influence its inner structure, including the distribution, size and shape of hydrogel pores. Similar effect was observed in ref. [32], where chitosan/poly (vinyl alcohol) hydrogel containing humic acids exhibited smaller-sized open cells with interconnected narrow spaces, which made it more attractive for solvent diffusion.

The second effect is possible interaction between the diffusion probes and HA and MHA during the transport of the probes through the hydrogel. At the beginning of the diffusion experiment, the passing probe front is in contact with hydrogel containing non-occupied binding sites and can interact with new free ones. When the front arrives at the end of the hydrogel layer and starts to permeate into the acceptor compartment, the reaction between the probe and active sites still proceeds but the interactions are in equilibrium. This means that the immobilization of the probe has the same rate as its liberation from binding sites. Therefore (in this stage), the probe is diffused through hydrogel which is saturated and (under ideal circumstances) such transport can be stationary with a constant diffusion rate characterized by the linear dependence of c_A_ on time (Figure 1).

If it is assumed that diffusion probes do not interact with pure AG hydrogel, the time lag determined for AG hydrogel (and the corresponding value of the apparent diffusion coefficient *D*_L_) are given only by the combination of the pore structure of hydrogels and the size and shape of the diffusion probes. A structure parameter (*μ*) can be defined by means of the porosity (*φ*) and tortuosity (*τ*) of the hydrogel [28,37,38,39,40,41,42,43]:*μ* = *φ*/*τ*.(3)

The value of *μ* can be determined as the ratio between the diffusion coefficient of the probe in AG hydrogel and the diffusion coefficient of the probe in water (*D*_0_). The values of *D*_0_ for metal ions are tabulated [48]: 1.43 × 10^−9^ m^2^·s^−1^ for Cu(II) and 1.32 × 10^−9^ m^2^·s^−1^ for Ni(II). The values of *D*_0_ for dyes were determined in our previous study [42]. They were extrapolated for 25 °C and used in this work as: 8.42 × 10^−10^ m^2^·s^−1^ for MB and 8.93 × 10^−10^ m^2^·s^−1^ for RH. These results are in agreement with values determined using other methods [48,49,50]. The ratios *D*_L_/*D*_0_ and *D*_E_/*D*_0_ are listed in Table 1, Table 2, Table 3 and Table 4. If we focus on the ratios obtained for pure AG hydrogel, we can state that their values are lower for the second stage of diffusion, when the probe is transported through the pores filled by its solution. The deceleration observed in the second stage can thus be connected with spatial conditions and the potential concentration dependence of the diffusion coefficient.

If AG hydrogels are enriched by HA and MHA, interactions between their binding sites and diffusion probes can proceed and influence probe transport. The difference between the diffusion coefficients obtained for pure AG hydrogel and hydrogels with incorporated HA and MHA can be considered as the result of interactions between probes and humic substances (and possible changes in hydrogel structure). The difference between HA and MHA can be seen in terms of the influence of carboxylic groups and those hydrophobized by methylation. Both diffusion coefficients (*D*_E_ and *D*_L_) for the diffusion of probes in AG-HA and AG-MHA hydrogels were determined by the structural characteristics of hydrogels, the potential concentration dependence of diffusion coefficient, and the influence of chemical reactions which can differ in first and second stages of the experiment.

As can be seen from Figure 2, the effect of humic substances incorporated into hydrogel was stronger in the case of the diffusion of dyes. Surprisingly, the transport was influenced more in the second stage of diffusion, when the system was equilibrated and the rates of direct and reverse reactions were the same. One explanation for this could be the sizes of dye molecules, which are bigger in comparison with metal ions. Therefore, after binding, they can reduce the effective pore diameters and suppress the diffusion. Another explanation could be the higher degree of immobilization of dyes, which means that they can be bound not only by negatively charged functional groups but also by hydroprobic structures within humic substances. In contrast, the transport of Ni(II) ions was practically unaffected by the presence of humic substances in the hydrogel. The diffusion of Cu(II) was affected mainly in the first stage of the experiment. This could be connected with the high affinity of copper to humic substances and the stability of the formed complexes, which resulted in low reaction rates after the achievement of reaction equilibrium.

### 2.2. Rheological Properties of Hydrogels

It is not easy to distinguish between the effect of structure and the effect of the reactivity of hydrogels on the transport of diffusion probes. In our previous works [41,42,43], it was assumed that potential changes in hydrogel structure caused by the addition of humic substances can be neglected. This study is focused more on the influence of humic acids on the structure of hydrogels and also on their transport properties. Rheological measurements provided information about the viscoelasticity of the studied hydrogels and changes in them caused by the addition of HA and MHA.

Figure 3 shows the dependencies of the storage and loss moduli on the strain at constant frequency. The storage modulus G’ proportional to the elastic component of the hydrogel decreased with strain. The strong decrease at the given strain means that the hydrogel could not resist the mechanical stress and that its structure was irreversibly damaged. The breakdown of the hydrogel network was connected with the correspondence of the maximum of the loss modulus G’’ with the viscous behavior of the hydrogel. In general, viscoelastic materials can relax at an applied stress by flowing. In the case of the hydrogels in our study, the flow achieved a maximum at the moment of the collapse of the hydrogel network. The maximum as well as the decreasing part of the G’curve was shifted to the left for AG-HA and AG-MHA hydrogels. This means that the AG hydrogel was more resistant to applied stress than hydrogels enriched with humic substances and that the networks of the AG-HA and AG-MHA hydrogels can easily collapse. Cross-over points (G’ = G’’) were determined at strains of 21%, 8%, and 5% and moduli of 1.8 kPa, 2.5 kPa, and 1.3 kPa for AG hydrogel, AG-HA hydrogel, and AG-MHA hydrogel, respectively. On the basis of these results, a strain of 0.05% was chosen for the measurement of the frequency dependencies of moduli. This strain lay in the regions of linear viscoelasticity of all the studied hydrogels and was lower than the strain causing the collapse of the hydrogel network.

In Figure 4, the frequency dependencies of the storage and loss moduli at low constant strain are shown. It was found that the extent of the elasticity (corresponding with the G’modulus) increased with increasing frequency. The G’’ modulus had a maximum at 0.03 Hz for AG-HA hydrogel and 0.02 Hz for AG-MHA hydrogel. No maximum was observed for pure AG hydrogel (without humic substances). The G’’ curves at higher frequencies had a similar character; the G’’ modulus decreased with increasing frequency and the flowing ability of hydrogels also decreased. A region of constant G’’ above a frequency of 1 Hz was observed for pure AG hydrogel.

The rheological data are summarized in Figure 5, where the dependence of the ratios between the loss modulus G’’ and storage modulus G’ on strain and frequency determined for all studied hydrogels are plotted. The G’’/G’ ratio represents a criterion of the “liquid behavior” of the studied samples. If the ratio is equal to one, the storage and loss moduli have the same values and the degrees of liquid and elastic extents are precisely balanced. Values higher than one denote that the sample behaves as a viscoelastic liquid. Liquid behavior was observed when the hydrogels were strained above a certain limit. In Figure 5a, this limit was around a strain of 16% for AG hydrogel, 11% for AG-HA hydrogel, and 6% for AG-MHA hydrogel. After crossing this limit, the G’’/G’ ratio increased strongly, the hydrogel network collapsed, and the samples behaved like liquids. The beginning of the increase in the G’’/G’ ratio corresponded with the maximum of the G’’ modulus in Figure 3b. Figure 5b shows the maxima of the G’’/G’ ratio for the AG-HA and AG-MHA hydrogels. These maxima represent the highest extent of the liquid behavior of the hydrogels, but the values are much lower than one and the extent of the elastic component predominated at all frequencies. The results show that the rheological properties of the hydrogels were altered by the addition of humic substances and that the effect was stronger for MHA. The behavior of hydrogels enriched with humic substances shifted towards that of viscoelastic liquids. This means that hydrogels containing humic substances had a lower ability to resist mechanical stresses, which can be connected with their higher permeability.

### 2.3. Relationships between Transport and Rheological Properties of Hydrogels

As described above, the final contents of humic substances in the hydrogels were very low (0.01% wt). However, their influence on the transport properties of hydrogels as well as on their viscoelasticity was not negligible. There could be two explanations for this result. The first is that the supramolecular structure of humic substances can strongly affect the “functioning” of hydrogels at very low contents. The increase in the permeability of Ni(II) ions can be caused by the change in the rheological properties of enriched hydrogels. Ni(II) ions are known as metals with very low affinity to humic substances and thus their transport through AG-HA and AG-MHA hydrogel should be affected only by changes in hydrogel structure. The addition of humic substances resulted in an increase in diffusion rate because of the shift in hydrogel viscoelasticity more towards the behavior of liquids and because of the lower resistance of hydrogels to mechanical stresses and the transport of probes. This positive effect on hydrogel permeability was eclipsed by a further property of humic substances, namely their reactivity. Cu(II) ions as well as both the dyes studied in this work have high affinity to humic substances and their interactions strongly affected their diffusion through hydrogels. Carboxylic groups play important roles in the interactions of copper with humic substances [35,36,37,38,39,40,41,42,43]. The selective blocking of these groups by methylation caused an increase in the *D*_L_ value determined for the first stage of the experiment, when the passing front of Cu(II) ions was in contact with the hydrogel containing non-occupied binding sites of humic substances. If the carboxylic groups were blocked by methylation, the interactions were less intensive and the transport through the hydrogel was accelerated. A similar effect was observed in the second stage of the experiment, but the interactions were in equilibrium and their influence was slightly suppressed. The situation surrounding the transport of dyes was more complex. They are bigger in comparison with metal ions, which resulted in a slower passage through the hydrogel. The “more liquid” character of hydrogels enriched with humic substances had a smaller effect than the interactions of dyes with humic substances. In contrast to metal ions, dyes, due to their cyclic condensed structures, can interact strongly with the hydrophobic domains of humic substances. Simultaneously, they can react with negatively charged functional groups (similarly to metal ions). The methylation of humic substances reduced their negative charge and supported their interactions with dyes via hydrophobic domains including methylated groups. This resulted in the further deceleration of the transport of dyes through AG-MHA hydrogel. This effect was less important when the dyes arrived at the end of the hydrogel layer and started to permeate into the acceptor compartment. The interactions were equilibrated in this stage of the experiment. The influence of the more liquid character of AG-MHA hydrogel can thus cause a gentle increase in the MB diffusion rate (and *D*_E_ value) in comparison with MB diffusion through AG-HA hydrogel. 

## 3. Materials and Methods

### 3.1. Chemicals

Agarose (AG; routine use class), CuCl_2_.2H_2_O (p.a.), NiCl_2_.6H_2_O (p.a.), methylene blue hydrate (MB; CI basic blue 9), and rhodamine 6G (RH; CI basic red 1) were purchased from Sigma-Aldrich (Prague, Czech Republic).

Humic acids (HA; Leonardite standard 1S104H) were purchased from the International Humic Substances Society (St. Paul, MN, USA). Their main characteristics such as elemental composition and the contents and properties of acidic functional groups can be found on the website of the International Humic Substances Society (http://humic-substances.org/). Their equilibrated models together with corresponding force-field parameters were provided by molecular dynamics simulations based on Vienna Soil Organic-Matter Modeler [51].

Methylated humic acids (MHA) were prepared from HA by means of the following procedure: 1 g of non-modified humic acids was mixed with 4 mL of CHCl_3_ and 2 mL of methanol. Then, 4 mL of 2 M solution of trimethylsilyldiazomethane in hexane (TMS-DM) were added. The mixture was stirred continuously for 2 h on a vortex. Subsequently, an additional 0.75 mL of TMS-DM was added. The obtained MHA were dried for 2 h under a nitrogen atmosphere and then overnight at 50 °C in an oven [39,40,43]. The composition and properties of HA and MHA are compared in [40]. 

### 3.2. Preparation of Hydrogels

The preparation of hydrogels was based on the thermo-reversible gelation of AG aqueous solution. AG was dissolved in deionized water or in an aqueous solution of HA or MHA, then heated (80 °C) and stirred to obtain a transparent solution, and finally sonicated to remove gasses. Afterwards, the solution was poured into a plastic ring mold fixed between two glass slides. The solution solidified gradually to hydrogel form by spontaneous cooling (to laboratory temperature). When the glasses were removed, a cylindrical hydrogel sample (40 mm in diameter and 5 mm thick) fixed in the plastic mold was obtained. AG hydrogels were prepared using 1% wt. AG solution. AG-HA (AG-MHA) hydrogels were prepared from 1% wt. AG solution containing 0.01% wt. of HA (MHA) [41,42,43]. 

### 3.3. Diffusion Experiments

The horizontal diffusion cell method was used in this study. The apparatus used was purchased from Permegear Inc. (Hellertown, PA, USA; for details, see our previous studies [41,43]). The hydrogel sample fixed in the plastic mold was placed between two apparatus compartments. The donor compartment was filled by aqueous solutions of dye or metal salt, the acceptor compartment by deionized water; thus, one circular surface of the hydrogel was in contact with the donor solution and second with water. A circulating water bath was used to maintain the temperature at 25 °C (the scheme of apparatus and experimental arrangement was published in [41]). Aqueous solutions of CuCl_2_, NiCl_2_, MB, and RH were used as the donor solutions. Their initial concentrations (*c*_D0_) were equal to 0.2 mol·dm^−3^ for metal salts and 0.01 g·dm^−3^ for dyes. The solutions in both compartments were stirred continuously (250 rpm) and the changes of in concentrations were monitored by means of a USB 2000+ fiber spectrometer (Ocean Optics, Inc., Largo, FL, USA) equipped with an optical dip probe.

### 3.4. Rheological Properties of Hydrogels

The hydrogels were sliced and the resulting cylindrical samples (1 mm in thickness) were used for rheological measurements. Each sample was placed between two titanium plates (40 mm in diameter) of an AR-G2 rheometer (TA Instruments, Ltd., New Castle, DE, USA) equipped with Rheology Advantage Instrument Control AR software. (v5.5.24, TA Instruments, Ltd., New Castle, DE, USA) Silicon oil was used to prevent drying of the hydrogels. The hydrogels were left to relax for 10 min before measurements were made. Measurements were performed at 25 ± 1 °C. Viscoelasticity was characterized by the storage modulus *G*’ (proportional to the extent of the elastic component) and loss modulus *G*’’ (proportional to the extent of the viscous component). All experiments were triplicated and average values are presented. 

## 4. Conclusions

It was concluded that the addition of HA and MHA can affect both the rheological behavior of AG hydrogels and their permeability for diffusion probes. The hydrogels enriched by supramolecular humic substances had a more liquid character and their resistance to mechanical stress was lower in comparison with pure AG hydrogel. This effect resulted in the faster transport of Ni(II) ions and an increase in the *D*_E_ value obtained for the diffusion of MB through AG-MHA hydrogel in the second stage of the experiment. In contrast, the interactions of HA and MHA with other probes resulted in the deceleration of their diffusion through the enriched hydrogels. In the case of Cu(II) ions, this effect was suppressed by the methylation of carboxylic groups, which have an important role in the interactions of metal ions with humic substances. The condensed cyclic structure of the dye probes supported their interactions with the hydrophobic domains of humic substances. 

## Figures and Tables

**Figure 1 molecules-24-01545-f001:**
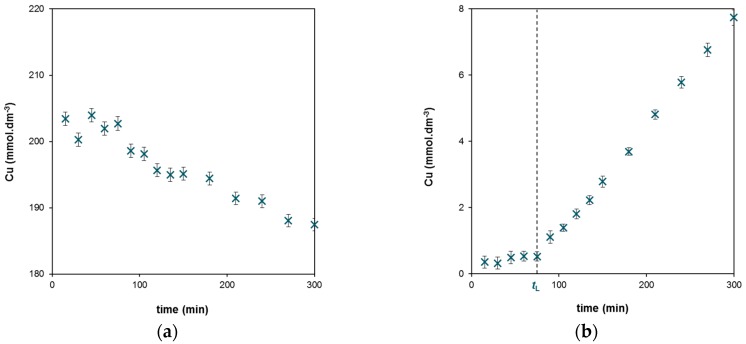
Experimental data obtained for copper(II) diffusing through hydrogel containing MHA: (**a**) Decrease in the concentration of Cu(II) ions in the donor compartment; (**b**) Increase in the concentration of Cu(II) ions in the acceptor compartment with marked time lag *t*_L_.

**Figure 2 molecules-24-01545-f002:**
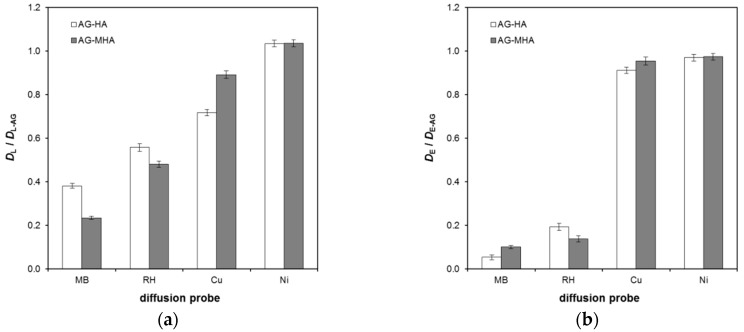
Ratios between the diffusion coefficients obtained for hydrogels enriched by humic substances (*D*_L_ and *D*_E_) and pure AG hydrogels (*D*_L_ and *D*_E_): (**a**) First stage of experiment—Equation (1); (**b**) Secondstage of experiment—Equation (2).

**Figure 3 molecules-24-01545-f003:**
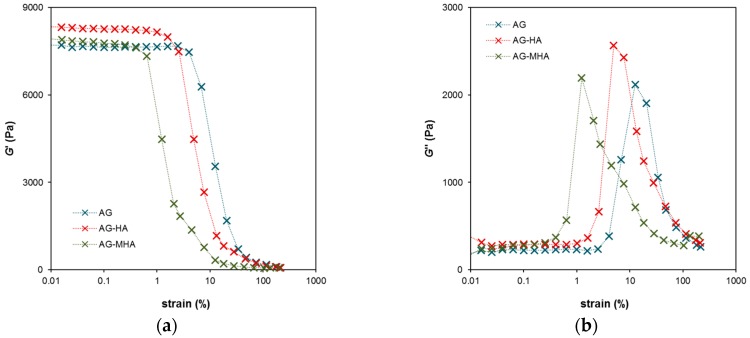
Viscoelastic properties of AG, AG-HA and AG-MHA hydrogels in dependence on strain at a frequency of 1 Hz: (**a**) Storage modulus *G*’, proportional to the extent of elastic component; (**b**) Loss modulus *G*’’, proportional to the extent of viscous component.

**Figure 4 molecules-24-01545-f004:**
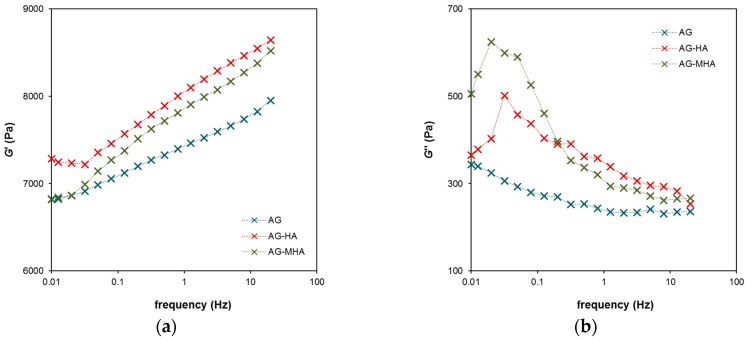
Viscoelastic properties of AG, AG-HA and AG-MHA hydrogels in dependence on frequency at a strain of 0.05%: (**a**) Storage modulus *G*’, proportional to the extent of elastic component; (**b**) Loss modulus *G*’’, proportional to the extent of viscous component.

**Figure 5 molecules-24-01545-f005:**
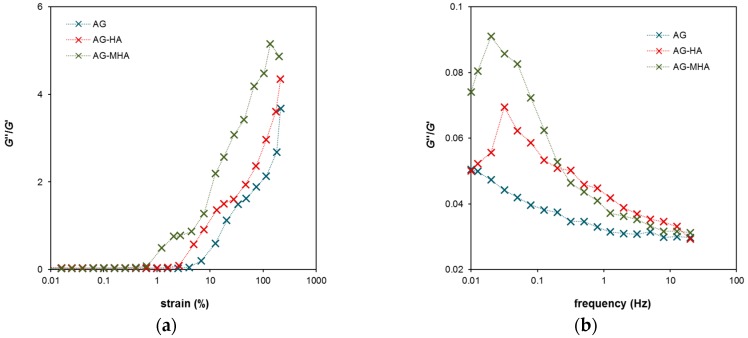
The ratios between the loss modulus G’’ and storage modulus G’ determined for AG, AG-HA and AG-MHA hydrogels: (**a**) The dependence on the strain at a constant frequency of 1 Hz; (**b**) The dependence on the frequency at a constant strain of 0.05%.

**Table 1 molecules-24-01545-t001:** Diffusion characteristics obtained for the transport of MB through hydrogels.

Hydrogel Sample	*t*_L_ (min)	*D*_L_ (10^−10^ m^2^·s^−1^)	*D*_E_ (10^−10^ m^2^·s^−1^)	*D*_L_/*D*_0_ (-)	*D*_E_/*D*_0_ (-)
AG	177	3.93	1.69	0.47	0.20
AG-HA	464	1.50	0.12	0.18	0.01
AG-MHA	774	0.92	0.17	0.11	0.02

**Table 2 molecules-24-01545-t002:** Diffusion characteristics obtained for the transport of RH through hydrogels.

Hydrogel Sample	*t*_L_ (min)	*D*_L_ (10^−10^ m^2^·s^−1^)	*D*_E_ (10^−10^ m^2^·s^−1^)	*D*_L_/*D*_0_ (-)	*D*_E_/*D*_0_ (-)
AG	242	2.86	2.01	0.32	0.23
AG-HA	436	1.60	0.39	0.18	0.04
AG-MHA	516	1.38	0.28	0.15	0.03

**Table 3 molecules-24-01545-t003:** Diffusion characteristics obtained for the transport of Cu(II) through hydrogels.

Hydrogel Sample	*t*_L_ (min)	*D*_L_ (10^−10^ m^2^·s^−1^)	*D*_E_ (10^−10^ m^2^·s^−1^)	*D*_L_/*D*_0_ (-)	*D*_E_/*D*_0_ (-)
AG	69	10.13	8.63	0.71	0.60
AG-HA	95	7.28	7.87	0.51	0.55
AG-MHA	77	9.03	8.23	0.63	0.58

**Table 4 molecules-24-01545-t004:** Diffusion characteristics obtained for the transport of Ni(II) through hydrogels.

Hydrogel Sample.	*t*_L_ (min)	*D*_L_ (10^−10^ m^2^·s^−1^)	*D*_E_ (10^−10^ m^2^·s^−1^)	*D*_L_/*D*_0_ (-)	*D*_E_/*D*_0_ (-)
AG	70	9.87	2.36	0.75	0.18
AG-HA	57	10.21	2.28	0.77	0.17
AG-MHA	57	10.22	2.29	0.77	0.17

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
