# Peer review of "How Humic Acids Affect the Rheological and Transport Properties of Hydrogels"

_molecules, 2019, doi:10.3390/molecules24081545_

Round 1

Reviewer 1 Report

The authors proposed the present study as a continuation of their previous efforts to elucidate the effect of conformational arrangement of humic acids supramolecular structures on their interaction with other substances. They prepared agarose hydrogels in the presence of two types of humic acids: the protonated H-form of humic acids and humic acids having the carboxylic functions blocked by methylation. The transport of metal ions and organic dyes through the resulted agarose hydrogels, enriched by humic acids, was monitored along with the change in the viscoelastic properties. Although the final contents of humic substances in the hydrogels were very low, their influence on the transport and rheological properties was significant. This effect resulted infaster transport of Ni(II) ions and in deceleration of the diffusion of the organic dyes and Cu(II) ions. The hydrogels enriched by supramolecular humic substances had a more liquid character and a lower resistance to mechanical stress as compared with the pure agarose hydrogel.

Reccomandations:

1. Although the introduction seemed rather long to me at first glance, it may still be accepted due to the useful information that it provides.

2. The Conclusions section is also too long. I would recommend that most of the comments from Conclusions to be moved to Results and Discussion section. The Conclusions section should contain only a few paragraphs that briefly outline the significant outcomes of the study.

I recommend publication of the article after minor revision.

Author Response

1. Although the introduction seemed rather long to me at first glance, it may still be accepted due to the useful information that it provides.

Introduction was reduced as was required by reviewer 2. Changes are in red.

2. The Conclusions section is also too long. I would recommend that most of the comments from Conclusions to be moved to Results and Discussion section. The Conclusions section should contain only a few paragraphs that briefly outline the significant outcomes of the study.

Conclusions section was reduced, some comments was moved to results and Discussion section. Changes are in red.

Reviewer 2 Report

Minor changes are required in respect of significance of the work and explanations should be given more precisely.

Specific comments

1.       The introduction section is divided into 6 paragraphs and it seems very lengthy. Authors are suggested to concise the introduction section with 4 paragraphs maximum.

2.       Recently published papers should be added more in the manuscript.

3.       Authors are suggested not to use active voice with first person plural number. Please replace all those sentences with passive voice and no use of any form of pronouns.

4.       The significance part is critical and it should be more clearly explained.

5.       The relation between rheological and transport properties of hydrogels containing humic substances needs more clarifications and especially Figs 4 and Figs. 5 should be redrawn with more data points .

6.       Conclusion part should be more descriptive and easily understandable.

Author Response

1.       The introduction section is divided into 6 paragraphs and it seems very lengthy. Authors are suggested to concise the introduction section with 4 paragraphs maximum.

Introduction was reduced, it has 4 paragraphs. Changes are in red.

2.       Recently published papers should be added more in the manuscript.

They were added. New references are in red.

3.       Authors are suggested not to use active voice with first person plural number. Please replace all those sentences with passive voice and no use of any form of pronouns.

Whole manuscript was revised. Sentences are written in passive voice. Changes are in red.

4.       The significance part is critical and it should be more clearly explained.

This part was rewritten. Conclusions section was reduced, some comments was moved to results and Discussion section. Changes are in red.

5.       The relation between rheological and transport properties of hydrogels containing humic substances needs more clarifications and especially Figs 4 and Figs. 5 should be redrawn with more data points .

In this moment, it is not possible to redrawn Figs 4 and 5, because the studied hydrogels were consumed. The preparation of new hydrogels requires new purchase of humic acids from the International Humic Substances Society and their new methylation.

The discussion about the relation between rheological and transport properties of hydrogels containing humic substances was changed in order to achieve better clarification. Changes are in red.

6.       Conclusion part should be more descriptive and easily understandable.

Conclusion section was changed and reduced, some comments was moved to results and Discussion section. Changes are in red.